# Soft Sampling for Efficient Training of Deep Neural Networks on Massive Data

## Abstract

We investigate soft sampling which is a simple yet effective approach for efficient training of large-scale deep neural network models when dealing with massive data. Soft sampling selects a subset uniformly at random with replacement from the full data set in each epoch. First, we derive a theoretical convergence guarantee for soft sampling on non-convex objective functions and give the convergence rate. Next, we analyze the data coverage and occupancy properties of soft sampling from the perspective of the coupon collector's problem. And finally, we evaluate soft sampling on various machine learning tasks using various network architectures and demonstrate its effectiveness. Compared to existing coreset-based data selection methods, soft sampling offers a better accuracy-efficiency trade-off. Especially on real-world industrial scale data sets, soft sampling can achieve significant speedup and competitive performance with almost no additional computing cost.

## 1 Introduction

Deep learning (LeCun et al., 2015) has made great progress in a broad variety of domains in recent years (Silver et al., 2016; Esteva et al., 2017; Saon et al., 2017; Xiong et al., 2017). The high performance of deep neural network models having huge numbers of parameters relies on large amounts of training data (Brown et al., 2020; Parthasarathi et al., 2019; Chowdhery et al., 2022). This comes with a cost of long training time and demands substantial computing and storage resources. High computational complexity sometimes becomes a barrier to the hyper-parameter tuning and model validation steps that are crucial for real-world deployments. In this situation, data selection is often used to select a representative subset of the entire training data to speed up the training while maintaining decent model performance.

Subset selection has been shown to be an effective approach to alleviating the computational cost in large scale machine learning (Mirzasoleiman et al., 2020a; Borsos et al., 2021; Kowal, 2022; Guo et al., 2022). It is also used in distributed training to reduce the communication cost (Reddi et al., 2015) and active learning to create compact sets for human labeling (Hakkani-Tur et al., 2002; Tur et al., 2003; Kaushal et al., 2019; Coleman et al., 2020). Usually a subset is selected based on some criterion such that the performance of a model trained on the subset is comparable to one trained on the whole dataset, but with much less data and computing efforts. A variety of criteria have been introduced in various applications in the literature. For instance, diversity reward is used in (Lin & Bilmes, 2011) for document summarization and in (Kaushal et al., 2019) for computer vision (CV). Text similarity and saturated coverage are used in (Wei et al., 2013) to select acoustic data for automatic speech recognition (ASR). The maximum entropy principle is applied in (Wu et al., 2007; Yu et al., 2009) to select an informative data subset. Confidence scores are used in (Hakkani-Tur et al., 2002; Tur et al., 2003) based on a well-trained model to select a subset with highest uncertainty for labeling for active learning. In (Sivasubramanian et al., 2021) error bounds on the validation set are taken into account when selecting a data subset for $\ell_2$ regularized regression problems for better model generalization. In (Mirzasoleiman et al., 2020a; Killamsetty et al., 2021a) subsets are selected to closely approximate the full gradient for training machine learning models using incremental gradient methods.

The construction of an optimal subset is combinatorial and NP-hard in principle. In (Wei et al., 2015; 2014b;a; Kirchhoff & Bilmes, 2014; Killamsetty et al., 2021b) subsets are selected leveraging submodular functions with diminishing returns where the subset selection can be formulated as

constrained submodular cover optimization (Fujishige, 1991). Subset selection is also viewed as summarizing the full data set using a coreset (e.g. weighted subset samples) in (Mirzasoleiman et al., 2020a;b; Reddi et al., 2015; Coleman et al., 2020; Killamsetty et al., 2021a). Most of the subset construction algorithms are greedy algorithms which are computationally efficient, and some of them can provide provable approximation guarantees compared to the solution on the full data set. For many of the existing data selection approaches, the selection is a hard selection where a subset of the full data is selected and models are trained on this constant subset of data while the samples outside the subset are totally discarded (Wu et al., 2007; Lin & Bilmes, 2011; Wei et al., 2014b). Furthermore, to reduce the cost of data selection, an additional light-weight proxy model is introduced for selecting subsets in a family of so-called selection via proxy (SVP) methods (Coleman et al., 2020; Sachdeva et al., 2021). However, even with greedy algorithms which are relatively efficient in constructing subsets or selection via proxy, many of the existing data selection techniques still suffer from scaling issues when dealing with large amounts of data and models of large capacity due to demanding processing time and memory requirements (Wei et al., 2014a; Mirzasoleiman et al., 2020a).

In this paper we propose soft sampling, a simple but effective approach to training models with reduced data size for efficiency. Soft sampling selects uniformly at random with replacement a subset from the full data set for each training epoch, so every data sample can be sampled with non-zero probability. The selection of data is agnostic to loss functions and models. Compared to deterministic loss/cost function based data selection methods, soft sampling is significantly faster without requiring additional memory, which makes it very suitable for training deep neural networks using incremental gradient techniques such as stochastic gradient descent (SGD) and its variants. Randomized subset selection has been presented in the literature (Pooladzandi et al., 2022; Killamsetty et al., 2021a; Guo et al., 2022), where it is mostly treated as an underperforming baseline. It is either compared with coreset selection methods on small datasets with a very low data selection percentage (e.g 1%) or it is not investigated at its full strength when the comparative study is made with other subset selection techniques. In this work we assess random subset selection as a low-cost data selection approach that is very computationally efficient when training deep models with a large number of parameters on large scale datasets. We study this random subset selection approach both theoretically and practically. We show that soft sampling is guaranteed to converge and give its convergence rate. We also analyze its statistical properties on sample coverage and occupancy. Experiments are carried out to extensively evaluate its effectiveness on a variety of datasets from image classification and speech recognition. We show that soft sampling can obtain competitive or superior performance compared with some existing high-performance data selection approaches while being much more efficient in speed and memory usage.

## 2 RELATED WORK

Subset selection is cast as submodular optimization in (Lin & Bilmes, 2009; 2011; Wei et al., 2013; 2014b;a; 2015; Kirchhoff & Bilmes, 2014; Mirzasoleiman et al., 2015) where submodular functions are defined on discrete sets and optimized under constraints (e.g. cardinality of the selected subset). Submodular optimization based subset selection is mathematically rigorous, as under mild conditions a simple greedy implementation is theoretically guaranteed to be only a constant fraction away from the optimal solution. However, despite the availability of a rich class of functions, suitable submodular functions still need to be carefully chosen and tailored to the problem under investigation given the computational complexity and scale of the data. Furthermore, once the subset is selected, it is usually fixed throughout the training regardless of the iteratively updated model.

Coreset algorithms have been explored in (Mirzasoleiman et al., 2020a; Killamsetty et al., 2021b;a; Pooladzandi et al., 2022) where weighted subsets are selected to summarize some desired properties of the full data for efficient training. GLISTER, proposed in (Killamsetty et al., 2021b), selects a coreset that maximizes the log-likelihood on a validation set. CRAIG in (Mirzasoleiman et al., 2020a) and GRAD-MATCH in (Killamsetty et al., 2021a) each find a coreset that closely approximates the full gradient. ADACORE in (Pooladzandi et al., 2022) extracts a coreset that dynamically approximates the curvature of the loss function based on the Hessian matrix. CRAIG, GRAD-MATCH and ADACORE are all adaptive methods which are shown to achieve superior performance over a fixed subset. ADACORE relies on second-order statistics which are more computationally demanding, while CRAIG and GRAD-MATCH search for first-order coresets which are computationally more efficient. In this work, we compare the performance on the accuracy-efficiency trade-off between soft

sampling and GRAD-MATCH (Killamsetty et al., 2021a). GRAD-MATCH is a first-order coreset selection approach to selecting coresets to approximate the full gradient. The selection is carried out using an efficient orthogonal matching pursuit (OMP) algorithm. We choose GRAD-MATCH as a baseline because it is a representative coreset selection approach and has been shown to outperform numerous existing high-performing subset selection techniques such as CRAIG and GLISTER in (Killamsetty et al., 2021a).

## 3 SOFT SAMPLING

Let $\mathcal{X}$ denote the input space and $\mathcal{Y}$ the output space. The goal of machine learning is to estimate a function $h$ that maps from the input to the output

$$h(x; w) : \mathcal{X} \to \mathcal{Y} \tag{1}$$

where $x \in \mathcal{X}$ and $h$ belongs to a family of functions parameterized by $w \in \mathbb{R}^d$. A loss function $f(h(x; w), y)$ is defined on $\mathcal{X} \times \mathcal{Y}$ to measure the closeness between the prediction $h(x; w)$ and the output $y \in \mathcal{Y}$. A risk function $\mathcal{L}(w)$ is defined as the expected loss over the underlying joint distribution $p(x, y)$:

$$\mathcal{L}(w) = \mathbb{E}_{(x,y)}[f(h(x; w), y)]. \tag{2}$$

We want to find parameters $w$ that minimize $\mathcal{L}(w)$

$$w^* = \arg\min_w \mathcal{L}(w). \tag{3}$$

In practice, we only have access to a training set $\mathcal{G}$ of $n$ data samples $\mathcal{G} = \{(x_i, y_i)\}_{i=1}^n \in \mathcal{X} \times \mathcal{Y}$ where $|\mathcal{G}| = n$ and the following empirical risk is minimized

$$\mathcal{L}_{\mathcal{G}}(w) = \frac{1}{|\mathcal{G}|} \sum_{i \in \mathcal{G}} f(h(x_i; w), y_i). \tag{4}$$

Incremental gradient methods such as SGD (Bottou et al., 2016; Bottou & Bousquet, 2007) and its variants (Kingma & Ba, 2015; Nesterov, 1983) have been the dominant approach in solving this optimization problem where at iteration $l$ a sample $(x_{i_l}, y_{i_l})$, $i_l \in \{1, \cdots, n\}$, is drawn at random from $\mathcal{G}$ and its stochastic gradient $\widehat{\nabla} f_{i_l}$ is then used to update $w$ with an appropriate stepsize $\alpha > 0$:

$$w_{l+1} = w_l - \alpha \widehat{\nabla} f_{i_l}(w_l). \tag{5}$$

When dealing with large scale machine learning, mini-batch based incremental gradient methods are commonly used for better trade-off between computing cost and approximation error (Bottou et al., 2016).

In case of a massive training set $\mathcal{G}$, a subset $\mathcal{V} \subset \mathcal{G}$ ($|\mathcal{V}| \ll |\mathcal{G}|$) is selected and the optimization is carried out only on $\mathcal{V}$ for computing efficiency. In a generic form, training after data selection can be given as

$$\mathcal{L}_{\mathcal{V}_k}(w) = \frac{1}{|\mathcal{V}_k|} \sum_{i \in \mathcal{V}_k} f(h(x_i; w), y_i) \tag{6}$$

where $\mathcal{V}_k$ is the subset selected for each epoch $k$ under some criterion (Wei et al., 2015; Mirzasoleiman et al., 2020a; Killamsetty et al., 2021b;a). $\mathcal{V}_k$ can be a constant subset in some works (Lin & Bilmes, 2011; Wei et al., 2014b).

In this paper we investigate soft sampling that efficiently trains machine learning models using randomized subsets. Suppose $|\mathcal{V}_k| = m$, for $k = 1, \cdots, K$. In each epoch $k$, instead of choosing a subset based on measures that are computationally demanding, we simply select a subset of size $m$ randomly from the ground set $\mathcal{G}$. Suppose $\Omega = \{\omega_1, \omega_2, \cdots\}$ are the $\binom{n}{m}$ subsets of size $m$. In each epoch, a subset is drawn with replacement from $\Omega$ with an equal probability to be used in the optimization of Eq.6. A detailed implementation is given in Algorithm 1.

---

**Algorithm 1** Training with soft sampling

---

$K \leftarrow$ Total number of epochs;
$n \leftarrow$ Total number of training samples;
$m \leftarrow$ Number of subset samples used in each epoch;
$\Psi \leftarrow$ SGD optimizer

Initialize $w_0$
Create $\Omega = \{\omega_1, \cdots, \omega_L\}$ consisting all subsets of size $m$ from ground set $\mathcal{G}$
**for** $k \leftarrow 1, \cdots, K$ **do**
    Select a subset $\omega_j$ uniformly at random with replacement from $\Omega$
    $\mathcal{V}_k \leftarrow \omega_j$
    $w_k \leftarrow \Psi(w_{k-1}, \mathcal{V}_k, \mathcal{L}_{\mathcal{V}_k})$

---

## 4 CONVERGENCE

We assume that (A1) the loss function is smooth and gradient Lipschitz continuous with constant $L$; and (A2) the gradient estimate is unbiased and has bounded variance, i.e., $\mathbb{E}[\widehat{\nabla} f_i(w)] = \nabla \mathcal{L}_{\mathcal{V}_k}(w)$, $\mathbb{E}\|\widehat{\nabla} f_i(w) - \nabla \mathcal{L}_{\mathcal{V}_k}(w)\|^2 \leq \sigma^2, \forall i \in \mathcal{V}_k$ and $\forall k$.

**Theorem 1.** *Suppose assumptions A1 and A2 hold and the iterates are generated by soft sampling. When the step size of Algorithm 1 satisfies $\alpha < 1/L$, we have*

$$\frac{1}{K} \sum_{k=1}^{K} \mathbb{E}\|\nabla \mathcal{L}_{\mathcal{G}}(w_k)\|^2 \leq \frac{2m(\mathcal{L}_{\mathcal{G}}(w_1) - \mathcal{L}(w^*))}{\alpha K} + \alpha m L \left(1 + \frac{m}{n}\right) \sigma^2 \tag{7}$$

*where the expectation is taken over all the randomness of the subset and data sample selection process. In addition, if $\mathcal{L}_{\mathcal{G}}$ satisfies the Polyak-Łojasiewicz inequality with $\mu > 0$, i.e., $\|\nabla \mathcal{L}_{\mathcal{G}}(w)\|^2 \geq 2\mu(\mathcal{L}_{\mathcal{G}}(w) - \mathcal{L}_{\mathcal{G}}(w^*))$, then*

$$\mathbb{E}\left[\mathcal{L}_{\mathcal{G}}(w_k) - \mathcal{L}_{\mathcal{G}}(w^*)\right] \leq (1 - \mu\alpha)^K \left(\mathcal{L}_{\mathcal{G}}(w_1) - \mathcal{L}_{\mathcal{G}}(w^*)\right) + 2\alpha\kappa m L \left(1 + \frac{m}{n}\right) \sigma^2, \tag{8}$$

*where condition number $\kappa := L/\mu$.*

*Remark.* Theorem 1 shows that when step size $\alpha \sim \mathcal{O}(1/\sqrt{K})$ the convergence rate of the proposed training scheme with soft sampling is $\mathcal{O}(1/\sqrt{K})$ (i.e., Algorithm 1 takes $\mathcal{O}(1/\epsilon^4)$ number of iterations to achieve an $\epsilon$-approximate first order stationary point of problem Eq. 3 under the empirical risk), which is the same as the standard SGD. Futher, when neural networks are overparametrized, the loss functions satisfy the Polyak-Łojasiewicz property (Jacot et al., 2018; Liu et al., 2022), therefore, Algorithm 1 with soft sampling is able to achieve the global optimal solution at the rate of $\mathcal{O}(1/K)$ when $\alpha \sim \mathcal{O}(1/K)$. Details of the proof are given in Appendix A.1.

## 5 SAMPLE COVERAGE AND OCCUPANCY

In soft sampling, subsets of samples are drawn with replacement from the ground set during training. In this section we investigate the data sample coverage and occupancy of soft sampling. Given $n$, the total number of samples in the ground set, and $m$, the number of samples in the subset used in each epoch, we are interested in answering the following questions:

**Coverage** How many samples in the ground set will we cover in training after $K$ epochs?

**Occupancy** How many epochs do we need in order to cover $s$ $(s \leq n)$ samples in the ground set?

The analysis can be cast into a balls-and-bins problem (Mitzenmacher & Upfal, 2005) where there are $n$ bins and every time $m$ balls are drawn and dropped into $m$ distinct bins. Each draw is independent and uniform at random. We want to analyze the distribution of non-empty bins after a number of draws. This is essentially a generalization of the coupon collector's problem (Mitzenmacher & Upfal, 2005) with group drawings (Stadje, 1990; Holst, 1986; Johnson & Kotz, 1977; David & Barton, 1962) where coupons come in groups of a constant size $m$ and all groups of coupons occur with equal probability.

## 5.1 COVERAGE

Let $\mathcal{S}$ denote the set of distinct training samples from the ground set after $K$ epochs of soft sampling and $|\mathcal{S}|$ denote the cardinality of $\mathcal{S}$. The distribution of $|\mathcal{S}|$ is given as (Stadje, 1990)

$$P(|\mathcal{S}| = l) = \binom{n}{l} \sum_{i=0}^{l} (-1)^i \binom{l}{i} \left[ \frac{\binom{l-i}{m}}{\binom{n}{m}} \right]^K , \quad l = 0, 1, \cdots, n. \tag{9}$$

Especially, when $l = n$, we have

$$P(|\mathcal{S}| = n) = \sum_{i=0}^{n} (-1)^i \binom{n}{i} \left[ \frac{\binom{n-i}{m}}{\binom{n}{m}} \right]^K \tag{10}$$

which is the probability of covering all the training samples after $K$ epochs of soft sampling.

From Eq.9, we have the expectation

$$\mathbb{E}[|\mathcal{S}|] = n \left[ 1 - \left( 1 - \frac{m}{n} \right)^K \right] \tag{11}$$

which is, on average, the number of covered training samples from the ground set after $K$ epochs. Table 1 shows the expected data coverage (in percentage) for various selection ratios ($\frac{m}{n}$) and numbers of epochs $K$.

| $m/n$ | $K=10$ | $K=20$ | $K=30$ |
|---|---|---|---|
| 5% | 40.1% | 64.2% | 78.5% |
| 10% | 65.1% | 87.8% | 95.8% |
| 20% | 89.3% | 98.8% | 99.9% |

Table 1: Expected data coverage in percentage of the ground set for various data selection ratios and numbers of epochs.

## 5.2 OCCUPANCY

Let $\bar{k}$ denote the number of draws (i.e. epochs) required to cover $s$ ($s \leq n$) samples in the ground set. The distribution of $\bar{k}$ is given as (Stadje, 1990)

$$P(\bar{k}) = \sum_{i=0}^{s-1} (-1)^{s-i+1} \binom{n}{i} \binom{n-i-1}{n-s} \frac{\binom{n}{m} - \binom{i}{m}}{\binom{n}{m}} \left( \frac{\binom{i}{m}}{\binom{n}{m}} \right)^{\bar{k}-1} , \quad \bar{k} = 1, 2, \cdots . \tag{12}$$

From Eq.12, we have its expectation

$$\mathbb{E}[\bar{k}] = \sum_{i=0}^{s-1} (-1)^{s-i+1} \binom{n}{i} \binom{n-i-1}{n-s} \frac{\binom{n}{m}}{\binom{n}{m} - \binom{i}{m}} . \tag{13}$$

When $s = n$, we have

$$P(\bar{k}) = \sum_{i=0}^{n-1} (-1)^{n-i+1} \binom{n}{i} \frac{\binom{n}{m} - \binom{i}{m}}{\binom{n}{m}} \left( \frac{\binom{i}{m}}{\binom{n}{m}} \right)^{\bar{k}-1} \tag{14}$$

and its expectation

$$\mathbb{E}[\bar{k}] = \sum_{i=0}^{n-1} (-1)^{n-i+1} \binom{n}{i} \frac{\binom{n}{m}}{\binom{n}{m} - \binom{i}{m}} \tag{15}$$

which is also given in (Ferrante & Saltalamacchia, 2014). Eq.15 gives the number of epochs required on average in order to cover the whole training ground set given the subset size $m$ and total sample size $n$.

In particular, when $m = 1$ we have

$$\mathbb{E}[\bar{k}] = \sum_{i=0}^{n-1} (-1)^{n-i+1} \binom{n}{i} \frac{n}{n-i} \tag{16}$$

$$\overset{j=n-i}{=} - \sum_{j=1}^{n} (-1)^j \binom{n}{j} \frac{n}{j} = n \left( - \sum_{j=1}^{n} (-1)^j \binom{n}{j} \frac{1}{j} \right)$$

$$= nH_n = n \log n + \mathcal{O}(n)$$

where $H_n = \sum_{i=1}^{n} \frac{1}{i}$ is the $n$th Harmonic number. Eq.16 is a well-known occupancy result for the classical coupon collector's problem (Mitzenmacher & Upfal, 2005).

## 6 EXPERIMENTS

We evaluate the accuracy-efficiency trade-off of soft sampling and compare with GRAD-MATCH, a high-performing coreset based subset selection approach, on image classification and automatic speech recognition (ASR) tasks. For the former we use the public CIFAR10 dataset. For the latter we use the public Librispeech dataset and an in-house Payload dataset. The Payload dataset is a real-world industrial scale dataset for training product-level ASR acoustic models. We used GRAD-MATCHPB-WARM (batch based GRAD-MATCH with a warm start) for the experiments because it gives the best performance compared to other GRAD-MATCH implementations in (Killamsetty et al., 2021a). In the CIFAR10 and Librispeech experiments soft sampling selects batches (similar to GRAD-MATCHPB), while in the Payload experiments soft sampling selects chunks of data due to the storage structure of this dataset and its massive size. In the experimental results, SS denotes soft sampling and GM denotes GRAD-MATCHPB-WARM. We use R to denote the selection interval where R1 stands for using different subsets for every epoch, which is the default setting for SS. R5 and R10 stand for selecting subsets every 5 and 10 epochs, respectively.

### 6.1 CIFAR10

The CIFAR10 dataset (Krizhevsky & Hinton, 2009) has 50,000 training images and 10,000 test images in 10 classes. We use the ResNet-18 model (He et al., 2015) with 11 million parameters. The batch size is 512 which is distributed to 4 P100 GPUs. The training ends after 320 epochs. A Nesterov accelerated SGD optimizer is used with a momentum of 0.9. The initial learning rate is 0.1 and it is annealed by 10x at the 160[th] epoch and the 240[th] epoch. The warm start of GM uses the 10[th] epoch of full data. The experimental results are given in Table 2.

### 6.2 LIBRISPEECH

The Librispeech dataset consists of 960 hours of 16kHz English audio from public domain audio books (Panayotov et al., 2015). There are about 30,000 utterances from 2338 speakers in the dataset with maximum duration of 35 seconds. Each utterance is converted to a sequence of frames every 10ms represented by a 40-dim logMel feature vector. We use the test-clean split to report word error rates (WERs). The acoustic model is a RNN-Transducer (RNN-T) (Graves, 2012). We use the standard training recipe from Speechbrain (Ravanelli et al., 2021). The transcription network has 2 convolutional layers followed by a 4-layer bi-directional LSTM (Hochreiter & Schmidhuber, 1997) and then 2 feed-forward layers. The prediction network is a single layer LSTM. The joint network projects the 1024-dimensional embeddings from the transcription and prediction networks to the output space of 1000 Byte-pair encoding units over the vocabulary. The decoding involves an external transformer language model trained on the Librispeech text. The RNN-T model has about 170 million parameters. The training uses an AdaDelta optimizer. The starting learning rate is 2.0 with an annealing factor of 0.8 for the relative improvement of 0.0025 on validation loss afterwards. The training is distributed on 2 A100 GPUs with a batch size of 24 utterances for 30 epochs. The warm start of GM uses the 2[nd] epoch of full data. The experimental results are given in Table 3.

## 6.3 PAYLOAD

The Payload dataset consists of 56,300 hours of English spontaneous speech data after data augmentation. Utterances are collected from real-world ASR services. The sampling rate is 8KHz. The set contains 20.3 million utterances with an average length of 10 seconds. The shortest utterances are around 0.1 seconds while the longest ones are around 333 seconds. Each utterance is converted to a sequence of frames every 10ms, and every two frames are represented by a 240-dim feature vector (logMel acoustic features and their first and second order derivatives), which gives rise to 10.1 billion feature vectors for the full training set. There are 8 test sets (S1 to S8) varying in duration from 1.4–7.3 hours with an average of 3.2 hours. They represent a good coverage of application domains in model deployment. The acoustic model is also an RNN-T. It has 6 bi-directional LSTM layers in the transcription network with 1,280 cells in each layer (640 cells per direction). The prediction network is a single-layer uni-directional LSTM with 1024 cells. The outputs of the transcription network and the prediction network are projected down to a 256-dimensional latent space where they are combined by element-wise multiplication in the joint network. After a hyperbolic tangent nonlinearity followed by an affine transform, it connects to a softmax layer consisting of 46 output units which correspond to 45 characters and the null symbol. The model has 59 million parameters. The RNN-T models are trained using the AdamW optimizer. The learning rate starts at $5 \times 10^{-4}$ and is annealed by $\frac{1}{\sqrt{2}}$ every epoch after 7 epochs. The training ends after 20 epochs. The batch size is 256 utterances which are distributed to 32 V100 GPUs. Since the dataset is large (2.4TB disk space for feature storage), it is divided into 320 chunks. The training is conducted sequentially by chunks. In each chunk the utterances are organized in a sorted order. This amounts to a curriculum learning strategy where it starts with short utterances to stabilize the training early on before gradually increasing to difficult longer utterances. SS is carried out by randomly selecting a subset of chunks. GM selects a batch subset across all chunks. The reason is that if GM selects entire chunks as SS it will significantly sacrifice the representative nature of a coreset. Furthermore, even if GM selects entire chunks it still has to go through every chunk to compute the gradient matching criterion in order to select the best subset. The warm start uses the 1st epoch of full data. Experimental results are given in Table 4.

Table 2: Accuracy (Acc) and training time (hours) of SS and GM on CIFAR10 under various training configurations and percentage of data selection. R1 denotes selection interval is every epoch and R10 denotes selection interval is every 10 epochs.

|  | SS_R1 | | SS_R10 | | GM_R1 | | GM_R10 | |
| --- | --- | --- | --- | --- | --- | --- | --- | --- |
|  | Acc | time | Acc | time | Acc | time | Acc | time |
| 100% | 95.08 | 0.60h | 95.08 | 0.60h | 95.08 | 0.60h | 95.08 | 0.60h |
| 5% | 89.59 | 0.03h | 87.24 | 0.03h | 89.88 | 1.52h | 87.44 | 0.18h |
| 10% | 92.11 | 0.06h | 90.47 | 0.06h | 92.11 | 1.55h | 90.45 | 0.21h |
| 20% | 93.27 | 0.12h | 92.71 | 0.12h | 93.50 | 1.60h | 92.63 | 0.27h |
| 30% | 94.29 | 0.18h | 93.37 | 0.18h | 93.83 | 1.66h | 93.25 | 0.33h |

Table 3: Word error rate (WER) and training time (hours) of SS and GM on Librispeech under various training configurations and percentage of data selection. R1 denotes selection interval is every epoch and R5 denotes selection interval is every 5 epochs.

|  | SS_R1 | | SS_R5 | | GM_R1 | | GM_R5 | |
| --- | --- | --- | --- | --- | --- | --- | --- | --- |
|  | WER | time | WER | time | WER | time | WER | time |
| 100% | 4.21 | 103.2h | 4.21 | 103.2h | 4.21 | 103.2h | 4.21 | 103.2h |
| 1% | 6.95 | 8.0h | 7.12 | 8.0h | 7.09 | 55.3h | 7.10 | 16.4h |
| 5% | 6.02 | 11.7h | 6.35 | 11.7h | 6.39 | 60.2h | 6.41 | 20.9h |
| 10% | 5.65 | 17.0h | 5.87 | 16.9h | 5.63 | 64.1h | 5.71 | 27.4h |
| 20% | 4.76 | 27.9h | 5.08 | 27.8h | 4.95 | 73.6h | 5.01 | 35.5h |
| 30% | 4.48 | 36.5h | 4.62 | 36.6h | 4.55 | 84.2h | 4.58 | 46.6h |

From Tables 2, 3 and 4, it can be observed that SS has a better accuracy-efficiency trade-off compared to GM considering recognition accuracy and training time. SS outperforms GM in most cases.

Table 4: Word error rate (WER) and training time (hours) of SS and GM on Payload under various training configurations and percentage of data selection. R1 denotes selection interval is every epoch and R10 denotes selection interval is every 10 epochs. In SS_R0 a random subset is selected and fixed for the training. In SS_R1_nw models are trained without warm start. Note that since SS is carried out at the chunk level while GM has to be carried out at the batch level, there is extra data loading time in GM. It takes about 42 seconds to load a chunk and 3.73 hours to load in all 320 chunks. That amounts to 74.6 hours for 20 epochs in the training.

|  |  | S1 | S2 | S3 | S4 | S5 | S6 | S7 | S8 | Avg. | Time |
|---|---|---|---|---|---|---|---|---|---|---|---|
|  | 100% | 6.2 | 9.7 | 6.3 | 22.6 | 16.5 | 25.3 | 16.3 | 29.0 | 16.49 | 426.7h |
| 5% | SS_R0 | 9.7 | 14.5 | 10.4 | 26.8 | 21.2 | 24.3 | 19.0 | 34.4 | 20.04 | 21.7h |
|  | SS_R1_nw | 9.5 | 14.6 | 9.9 | 26.4 | 21.3 | 24.7 | 19.0 | 33.8 | 19.90 | 22.1h |
|  | SS_R1 | 7.5 | 12.0 | 8.0 | 23.7 | 18.9 | 24.1 | 17.6 | 30.6 | 17.80 | 42.3h |
|  | GM_R10 | 7.5 | 12.1 | 7.4 | 23.8 | 18.7 | 25.7 | 17.6 | 31.2 | 18.00 | 128.3h |
| 10% | SS_R0 | 8.2 | 12.6 | 8.3 | 24.5 | 19.7 | 24.0 | 17.7 | 32.3 | 18.41 | 41.6h |
|  | SS_R1_nw | 7.8 | 12.0 | 8.6 | 24.3 | 19.1 | 22.3 | 17.2 | 31.7 | 17.88 | 40.9h |
|  | SS_R1 | 7.1 | 11.4 | 7.8 | 23.5 | 18.3 | 24.6 | 17.5 | 30.3 | 17.56 | 59.8h |
|  | GM_R10 | 7.1 | 11.4 | 7.5 | 23.5 | 18.2 | 26.7 | 18.2 | 30.8 | 17.93 | 192.9h |
| 20% | SS_R0 | 7.2 | 11.1 | 7.9 | 23.8 | 18.0 | 25.0 | 17.7 | 30.5 | 17.65 | 89.0h |
|  | SS_R1_nw | 7.0 | 11.0 | 7.1 | 23.5 | 18.1 | 23.9 | 16.6 | 29.9 | 17.14 | 89.6h |
|  | SS_R1 | 6.8 | 10.7 | 7.2 | 23.0 | 17.7 | 24.9 | 17.4 | 29.9 | 17.20 | 106.4h |
|  | GM_R10 | 6.9 | 11.0 | 7.2 | 23.2 | 17.5 | 26.9 | 17.8 | 30.3 | 17.60 | 314.5h |

GM only outperforms SS in cases when the selected subset is very small (e.g. 1% or 5%) and the two have the same selection interval (e.g. both with R1, R5 or R10). Even in this case, the difference of recognition accuracy between the two is not significant. However, SS has the same computational cost regardless of selection interval while GM has increasing computational cost when the selection interval is reduced. Taking that into account, SS can still outperform GM in the small subset conditions. This can be observed in Table 2 where for the 5% case SS_R1 has a better accuracy (89.59%) and shorter training time (0.03h) than GM_R10 (87.44% and 0.18h), in Table 3 where for the 1% case SS_R1 has a better WER (6.95%) and shorter training time (8h) than GM_R5 (7.10% and 16.4h), in Table 4 where for the 5% case SS_R1 has a better WER (17.80%) and shorter training time (42.3h) than GM_R10 (18.00% and 128.3h).

The advantage of SS is apparent when the size of the dataset is large. The computational cost of data selection in GM becomes more demanding when dealing with a large scale training set. In the payload experiments, the data selection in GM takes about 23 hours, which is even longer than one SGD epoch using the full training data (about 21.3 hours).

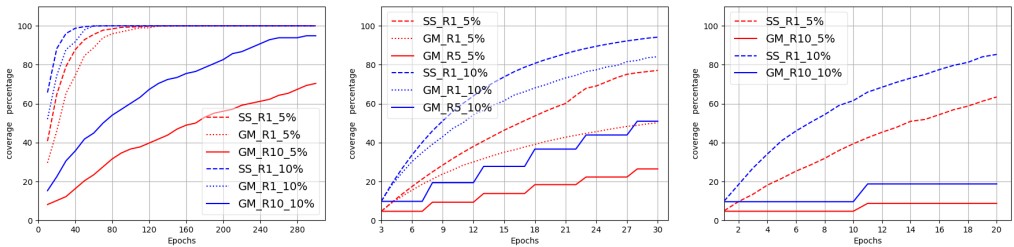

Figure 1: Percentage of data coverage using SS and GM for CIFAR10, Librispeech and Payload datasets when 5% and 10% subsets are selected.

Fig.1 demonstrates the percentage of data coverage using SS and GM when 5% and 10% of data are selected from the full data in each epoch with different selection intervals on the three datasets. It can be seen from the figure that the coverage of distinct data samples increases both under SS and GM. If the number of epochs goes higher (e.g. 320 epochs in CIFAR10), eventually both SS and GM will tend to achieve a very high coverage of distinct data samples. However, SS has an obvious higher

coverage rate than GM does when the number of epochs are not large (e.g. 30 epochs in Librispeech and 20 epochs in Payload). In addition, due to the computational cost, GM usually can not afford to make the subset adaptive every epoch which means its selection interval is typically larger than one. For example, the selection interval is 10 epochs in CIFAR10 and Payload and is 5 epochs in Librispeech in order to strike a reasonable balance between selection accuracy and computing efficiency. Under this condition, the coverage of distinct data samples using GM is much lower than that of SS. A higher data coverage in SS could benefit the training as the models learn from more data given the same computing budget. Also note that the practically observed sample coverage in Fig.1 is in line with the theoretical estimate in Table 1.

## 7 DISCUSSION

The coreset based data selection methods are typically resource and time demanding. GRAD-MATCH has to go through the full training set in order to compute the full gradient, requiring $\mathcal{O}(n)$ gradient evaluations. Furthermore, the greedy algorithm in OMP also requires $\mathcal{O}(nm)$ evaluations of the gains when selecting a data sample. When dealing with large models and massive data, the time and memory overhead could be prohibitive. Therefore the implementation of most coreset based data selection methods involves various approximations to improve efficiency. For instance, the gradient of the last layer is used to approximate the gradient of the whole model in the case of deep models, and the coreset selection is performed at the batch level instead of sample level. To guarantee a good subset selection at the start of training, a warm start is often used which requires a few SGD epochs using the full data. Despite an elegant theoretical guarantee under submodularity, the fast OMP implementation may give rise to sub-optimal solutions because the approximation error is dependent on $1 - \exp(-\lambda/(\lambda + k\nabla^2_{\max}))$. When $\lambda$ is large, the regularized problem is not the original one anymore. When $k$ is large, there is less theoretical benefit of selecting the subset. Compared to these first-order coresets (Mirzasoleiman et al., 2020a; Killamsetty et al., 2021a), second order coresets (Pooladzandi et al., 2022) may face even more severe issues in scaling.

Compared to the first and second order coreset based data selection, soft sampling incurs virtually zero time and memory cost in data selection. In addition, given the selection budget, soft sampling has more flexibility in choosing the selection granularity of subsets in accordance with the data structure, which is desirable when the training data is massive (e.g. the payload data). Although for a randomly selected subset in soft sampling the approximation error can not be guaranteed to be optimal under certain criteria (e.g full gradient approximation), soft sampling can offer frequently updated subsets across epochs that can provide a higher coverage of training data under the same per-epoch budget. This may help model generalization.

If only considering accuracy, coreset based data selection has advantages in that coresets are more representative of the full training set and they can give good results with lower data sample coverage compared to soft sampling, especially under a small selection budget. It is its computational complexity that makes it less efficient on massive training data. It should be noted that for coreset based data selection a trade-off can be made between time and resources. The data selection can rely on parallelization to significantly reduce the processing time, but it will meanwhile impose significant demands on CPU/GPU and memory usage.

## 8 CONCLUSION

In this paper we investigate soft sampling for efficient training of deep neural network models on large scale data. Soft sampling is computationally efficient with virtually no additional cost in data selection. Theoretically, we show that soft sampling has a convergence guarantee on non-convex objective functions and we provide the convergence rate. We also study the data coverage and occupancy properties of soft sampling. Practically, we compare soft sampling with GRAD-MATCH, a high-performing first-order coreset selection approach, on various datasets using various deep neural network models including an industrial scale ASR application. We show that soft sampling can provide a better accuracy-efficiency trade-off, which makes it very suitable for large scale training.

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

# A  APPENDIX

## A.1  PROOF OF THEOREM 1

Recall that $\mathcal{V}_k$ is the set of data samples randomly chosen from $[n]$, where $|\mathcal{V}_k| = m \ll n$. Also, let $\mathcal{F}_k^l = \{w_k^l, \ldots, w_1\}$ as filtration of the iterates generated by soft sampling with $w_k^1 = w_k, \forall k$. We denote $w_k^l$ as the iterate updated by the SGD optimizer $l$ times at the $k$ epoch. Then, the iterative algorithm can be written concisely as

$$w_k^{l+1} = w_k^l - \alpha \widehat{\nabla} f_i(w_k^l), \quad i \in \mathcal{V}_k. \tag{17}$$

Under the unbiasedness assumption of the gradient estimate, we know that

$$\mathbb{E}_i \left[ \widehat{\nabla} f_i(w_k^l) \right] = \widehat{\nabla} \mathcal{L}_{\mathcal{V}_k}(w_k^l), \forall i \in \mathcal{V}_k. \tag{18}$$

According to the gradient Lipschitz continuity of the objective function, we have

$$\mathbb{E} \left[ \mathcal{L}_{\mathcal{G}}(w_k^{l+1}) | \mathcal{F}_k^l \right]$$

$$\leq \mathcal{L}_{\mathcal{G}}(w_k^l) + \langle \nabla \mathcal{L}_{\mathcal{G}}(w_k^l), w_k^{l+1} - w_k^l \rangle + \frac{L}{2} \|w_k^{l+1} - w_k^l\|^2 \tag{19}$$

$$= \mathcal{L}_{\mathcal{G}}(w_k^l) - \alpha \langle \nabla \mathcal{L}_{\mathcal{G}}(w_k^l), \mathbb{E} \widehat{\nabla} f_i(w_k^l) | \mathcal{F}_k^l \rangle + \frac{\alpha^2 L}{2} \mathbb{E} \left[ \|\widehat{\nabla} f_i(w_k^l)\|^2 | \mathcal{F}_k^l \right] \tag{20}$$

$$\overset{(a)}{=} \mathcal{L}_{\mathcal{G}}(w_k^l) - \alpha \|\nabla \mathcal{L}_{\mathcal{G}}(w_k^l)\|^2 + \frac{\alpha^2 L}{2} \mathbb{E} \left[ \|\widehat{\nabla} f_i(w_k^l)\|^2 | \mathcal{F}_k^l \right] \tag{21}$$

$$\leq \mathcal{L}_{\mathcal{G}}(w_k^l) - \alpha \|\nabla \mathcal{L}_{\mathcal{G}}(w_k^l)\|^2 + \frac{\alpha^2 L}{2} \mathbb{E} \left[ \|\widehat{\nabla} f_i(w_k^l) - \nabla \mathcal{L}_{\mathcal{G}}(w_k^l)\|^2 | \mathcal{F}_k^l \right] \tag{22}$$

$$+ \frac{\alpha^2 L}{2} \left[ \|\mathbb{E} \widehat{\nabla} f_i(w_k^l)\|^2 | \mathcal{F}_k^l \right] \tag{23}$$

$$\overset{(b)}{\leq} \mathcal{L}_{\mathcal{G}}(w_k^l) - \alpha \left( 1 - \frac{\alpha L}{2} \right) \|\nabla \mathcal{L}_{\mathcal{G}}(w_k^l)\|^2 + \alpha^2 L \left( 1 + \frac{m}{n} \right) \sigma^2 \tag{24}$$

where $(a)$ is true because

$$\mathbb{E} \left[ \widehat{\nabla} f_i(w_k^l) | \mathcal{F}_k^l \right] = \mathbb{E}_{\mathcal{V}_k} \left[ \mathbb{E}_i \left[ \widehat{\nabla} f_i(w_k^l) | \mathcal{V}_k, \mathcal{F}_k^l \right] \right], \quad i \in \mathcal{V}_k \tag{25}$$

$$= \mathbb{E}_{\mathcal{V}_k} \left[ \widehat{\nabla} \mathcal{L}_{\mathcal{V}_k}(w_k^l) | \mathcal{F}_k^l \right] = \nabla \mathcal{L}_{\mathcal{G}}(w_k^l), \tag{26}$$

and $(b)$ follows due to

$$\mathbb{E} \left[ \|\widehat{\nabla} f_i(w_k^l) - \nabla \mathcal{L}_{\mathcal{G}}(w_k^l)\|^2 | \mathcal{F}_k^l \right] \quad i \in \mathcal{V}_k$$

$$= \mathbb{E} \left[ \|\widehat{\nabla} f_i(w_k^l) - \nabla \mathcal{L}_{\mathcal{V}_k}(w_k^l) + \nabla \mathcal{L}_{\mathcal{V}_k}(w_k^l) - \nabla \mathcal{L}_{\mathcal{G}}(w_k^l)\|^2 | \mathcal{F}_k^l \right] \tag{27}$$

$$\leq 2 \left( \sigma^2 + \frac{m}{n} \sigma^2 \right). \tag{28}$$

When $1 - \frac{\alpha L}{2} > 1/2$, i.e., $\alpha < \frac{1}{L}$, then, we have

$$\mathcal{L}_{\mathcal{G}}(w_k^{l+1}) \leq \mathcal{L}_{\mathcal{G}}(w_k^l) - \frac{\alpha}{2} \|\nabla \mathcal{L}_{\mathcal{G}}(w_k^l)\|^2 + \alpha^2 L \left( 1 + \frac{m}{n} \right) \sigma^2. \tag{29}$$

Taking the expectation over $\mathcal{F}_k^l$ and applying the telescoping sum over both $k$ and $l$ give

$$\frac{1}{K} \sum_{k=1}^{K} \frac{\alpha}{2} \mathbb{E} \|\nabla \mathcal{L}_{\mathcal{G}}(w_k)\|^2 \leq \frac{m \left( \mathcal{L}_{\mathcal{G}}(w_1) - \mathcal{L}_{\mathcal{G}}(w_K) \right)}{K} + \alpha^2 m L \left( 1 + \frac{m}{n} \right) \sigma^2. \tag{30}$$

Therefore, When $\alpha \sim \mathcal{O}(1/\sqrt{K})$, we have

$$\frac{1}{K} \sum_{k=1}^{K} \mathbb{E} \|\nabla \mathcal{L}_{\mathcal{G}}(w_k)\|^2 \leq \frac{2m(\mathcal{L}_{\mathcal{G}}(w_1) - \mathcal{L}_{\mathcal{G}}(w_K))}{\alpha K} + \alpha m L \left( 1 + \frac{m}{n} \right) \sigma^2, \tag{31}$$

resulting in the convergence rate of $\mathcal{O}(1/\sqrt{K})$. Alternatively, it implies that Algorithm 1 needs $\mathcal{O}(1/\epsilon^4)$ number of iterations to achieve an $\epsilon$-approximate first order stationary point (i.e., $\mathbb{E}\|\nabla\mathcal{L}_\mathcal{G}(w)\| \le \epsilon$). Applying the definition of $w^*$ gives the desired result.

When $\mathcal{L}_\mathcal{G}$ satisfies the Polyak-Łojasiewicz condition, then, from Eq. 29 we have

$$
\mathcal{L}_\mathcal{G}(w_k^{l+1}) - \mathcal{L}_\mathcal{G}(w^*)
$$

$$
\le \mathcal{L}_\mathcal{G}(w_k^l) - \mathcal{L}_\mathcal{G}(w^*) - \frac{\alpha}{2}\|\nabla\mathcal{L}_\mathcal{G}(w_k^l)\|^2 + \alpha^2 L \left(1 + \frac{m}{n}\right)\sigma^2 \tag{32}
$$

$$
\le (1 - \mu\alpha)\left(\mathcal{L}_\mathcal{G}(w_k^l) - \mathcal{L}_\mathcal{G}(w^*)\right) + \alpha^2 L \left(1 + \frac{m}{n}\right)\sigma^2 \tag{33}
$$

$$
\le (1 - \mu\alpha)^K \left(\mathcal{L}_\mathcal{G}(w_1) - \mathcal{L}_\mathcal{G}(w^*)\right) + \alpha^2 L \left(1 + \frac{m}{n}\right)\sigma^2 \sum_{j=1}^{mK-1}(1 - \mu\alpha)^j \tag{34}
$$

$$
\le (1 - \mu\alpha)^K \left(\mathcal{L}_\mathcal{G}(w_1) - \mathcal{L}_\mathcal{G}(w^*)\right) + 2\alpha\kappa m L \left(1 + \frac{m}{n}\right)\sigma^2, \quad \forall l, k \tag{35}
$$

which completes the proof.

