# OpenReview forum: "Soft Sampling for Efficient Training of Deep Neural Networks on Massive Data"
_ICLR.cc/2023/Conference — Submitted to ICLR 2023_

### Official Review · Reviewer_BYaz · 2022-10-24

**Confidence:** 4
**Correctness:** 4
**Technical Novelty And Significance:** 1
**Empirical Novelty And Significance:** 2
**Recommendation:** 3

**Clarity, Quality, Novelty And Reproducibility:**

Clarity: I believe the method is clear, though the name is confusing.

Quality: Experiments seem sound and show clear improvement on multiple datasets, but only one baseline is used. Admittedly, I did not verify the theoretical proof (I saw little point in doing so).

Novelty: I see no novelty in this method.

Reproducibility: the experiments section mentions the hyper-parameters and the GPU used. However, no code seems to be available.

**Strength And Weaknesses:**

Strengths:
- training time improvements on the baseline (GradMatch) are significant (e.g. when comparing GM_R1 and SS_R1 on CIFAR10)
- theoretical guarantees are provided

Weaknesses:
- the method name is very confusing. Looking at algorithm 1, I would simply call this method a variant of random sampling. I can understand that the "soft" part is trying to show that the data selection is not fixed (i.e. hard), so "soft random sampling" (with replacement) could be a better name
- given the above, I see no novelty in this method. Random sampling* is one of the first and most important baselines for data selection. As such, I'm surprised to see this being proposed as a novel method. The strength of this baseline has also been reported in multiple earlier works, for example, DeepCore (Chengcheng Guo et. al.) concludes: "... although various methods have advantages in certain experiment settings, random selection is still a strong baseline.". I believe this is a fatal flaw.


*: I understand that "random sampling" can refer to different but related methods: hard vs soft (or online vs offline) as well as with/without replacement.

**Summary Of The Paper:**

The authors propose a method called "Soft Sampling" which is a data selection method for training of deep learning models. They also show some theoretical convergence guarantees and experimentally show that their method achieves similar accuracy as GradMatch (a SoTA method) while using less resources (as measured by wall-clock training time). The datasets used are CIFAR10 (image classification), Librispeech (audio to text), and their in-house Payload dataset (audio to text).

The way I understand it, Soft Sampling simply selects a random subset with replacement given the budget at each epoch, as such the subset is not pre-determined/fixed.

**Summary Of The Review:**

Unless I'm missing something, which I'm hoping the authors can point out, the proposed method is not novel and has been one of the baselines that most data selection methods are always compared to. The strength of this baseline has also been observed many times before (e.g. DeepCore). As such, even though I believe the problem (data selection) to be highly important and knowing that I believe the method to be valuable, I must reject this paper.

---

> ### Author Response · Authors · 2022-11-17
> **response to reviewer BYaz**
>
> **Q1: the method name is very confusing. Looking at algorithm 1, I would simply call this method a variant of random sampling. I can understand that the "soft" part is trying to show that the data selection is not fixed (i.e. hard), so "soft random sampling" (with replacement) could be a better name.**
>
> **A1:** Thanks for the suggestion. We will consider it.
>
>
> **Q2: given the above, I see no novelty in this method. Random sampling is one of the first and most important baselines for data selection. As such, I'm surprised to see this being proposed as a novel method. The strength of this baseline has also been reported in multiple earlier works, for example, DeepCore (Chengcheng Guo et. al.) concludes: "... although various methods have advantages in certain experiment settings, random selection is still a strong baseline.". I believe this is a fatal flaw.**
>
> **A2:** Please kindly note that we do not intend to claim soft sampling as a brand new technique and we acknowledged in the paper that such randomized subset selection has been used in the literature.  Please refer to  the 2nd paragraph on page 2 of the paper where it also includes the one that you referenced here (DeepCore by C. Guo et. al.).   Instead, our contribution is the demonstration that, in the regime of large-scale training problems (here large means massive data), random sampling with only a small number of samples (i.e., soft version of the random sampling) is enough and even better in terms of both computational efficiency and generalization performance compared with the SOTA coreset methods, e.g., GRAD-MATCH, based on our theoretical analysis of convergence and data coverage. One of the novelties is that in this large-scale setting, soft sampling has already strikes a good tradeoff between the computational complexity and accuracy without loss of the convergence (due to sampling with replacement).
>
> Please also note that there are caveats in existing literature on randomized subset selection which need to pay careful attention.  It appears to us that most of the published randomized subset selection may not leverage its full strength for various reasons.  For instance, the DeepCore paper you mentioned in your review actually uses a fixed randomly selected subset to our understanding. In the last paragraph of Sec. 4.1, "For GradMatch, the experiment setting in the original paper is adaptive sampling, where subsets iteratively updated along with network training. Here, for a fair comparison, coresets are selected and then fixed for all training epochs." This actually corresponds to our SS_R0 setting in Table 4.  Also, in the GRAD-MATCH paper, random sets are compared but they are only updated in the same frequency as GRAD-MATCH.  This seems to be an unfair comparison as updated randomized subset  should be used in each epoch as it takes no extra cost.
>
> **Q3: Unless I'm missing something, which I'm hoping the authors can point out, the proposed method is not novel and has been one of the baselines that most data selection methods are always compared to. The strength of this baseline has also been observed many times before (e.g. DeepCore). As such, even though I believe the problem (data selection) to be highly important and knowing that I believe the method to be valuable, I must reject this paper.**
>
> **A3:** Please kindly note that we do not intend to claim soft sampling as a brand new technique and we acknowledged in the paper that such randomized subset selection has been used in the literature. Instead, what we want to argue is that randomized subset selection in most cases is treated as an under-performing and under-rated baseline which is used to compare with other coreset selection techniques.  However, when it comes to large scale training with massive data, random sampling is actually a high-performing approach in terms of accuracy, speed and resource. Therefore, we want to give a thorough study on such randomized subset selection including its convergence property and it sampling dynamics which we don't see in existing literature. Compared to coreset techniques, soft sampling has great advantages in data coverage given the same computing budget constraints.  This is the novelty that we want to show readers.

---

### Official Review · Reviewer_BCTj · 2022-10-24

**Confidence:** 4
**Clarity, Quality, Novelty And Reproducibility:** Clarity in paper is very good. Howeve…
**Correctness:** 3
**Technical Novelty And Significance:** 1
**Empirical Novelty And Significance:** 1
**Recommendation:** 3

**Strength And Weaknesses:**

The paper is well written. However, it has several weaknesses. Please see the points 2, 3, 4, 5

**Summary Of The Paper:**


1. The paper argues in favor of random sampling (termed as soft sampling) over other methods - specifically coreset based. The argument is well understood and to summarize
   a. The advantage of coresets in accuracy over random sampling is limited in some (maybe authors want to argue over many / most? But the evidence is provided on 3 datasets, 1 of which is very large) large scale datasets.
   b.  On the other hand, the computational cost of coresets is usually very high which is well known and accepted.
To extend the argument, the time spent in coreset computation can actually be used to train models for longer times with random sampling

2. The analysis seems fairly straightforward which is not a bad thing in my opinion. However, it definitely requires more scrutiny. Specifically, how is the convergence analysis different from mini-batch SGD analysis with bounded variance and smooth and lipschitz continuous ? More specifically, what new thing do we learn here ? Also, sample coverage and occupancy computations are important to understand the dynamics of random sampling in the context of training. However, to get fair understanding of the contribution, you might want to compare against a mini-batch SGD and talk about number of batches vs occupancy / coverage. (instead of epochs)

3. In the results, authors use same number of epochs across different methods. However, when sampling, the number of epochs really loose their meaning. Instead, we can keep number of iterations the same . Because it is difficult to compare between, say, 10% and 20% sampling rate if the 20% is runs twice as many iterations.

4. The argument that random sampling is competent for a lot of cases is a well accepted belief in the community. I am not sure if there has been a extensive study on the topic though. However, it should be noted that there will be cases where coresets will strongly outperform random sampling w.r.t accuracy and showing that random sampling beats coresets on a handful of datasets does not support using random sampling is better than coresets for training. In order to make a strong argument, i would suggest the following two directions -

a. Use a lot of real datasets from different domains with learning and do an extensive study confirming what people believe. Having such a study would be very useful.
b. Else, maybe you want to make an argument in favor of random sampling even in adversarial cases. In such a case, hand design data distributions where random sampling is bad (so in terms of #iterations, coresets beat random sampling) . But in training scenarios, using the advantage of fast sampling random sampling is competent with coresets in terms of #seconds to converge.

5. Also, it is not clear what the novelty of the paper is specifically because random sampling for training is the norm. It would be useful if authors would elaborate on the novelty. One interesting direction that i see in random sampling itself is this trade-off between soft-hard sampling. For example, the following two are natural extremes in random sampling,
[softest] you pick each sample with replacement and form batches from these samples
[hardest] you shuffle the entire dataset and make batches from this shuffled dataset. This is also what is current practice.
Your approach is then somewhere in the middle, where you select a subset (which obviously has no repetitions) and then make batches from the subset. However the two different subset selections can have overlaps.

You can analyse effect of this granularity that you introduce w.r.t occupancy, coverage and variance of gradient estimate for convergence (in theory and simulations) and in training convergence performance (# iterations required to converge).



**Summary Of The Review:**

The major concern is that the paper does not provide clear evidence to the effect that random sampling should be always preferred in training neural networks.  See points above in summary

---

> ### Author Response · Authors · 2022-11-17
> **response to reviewer BCTj**
>
> First of all, we would like to thank the reviewer for his/her insightful comments, which we believe would be very helpful to improve the quality of the work.
>
> **Q1: The analysis seems fairly straightforward which is not a bad thing in my opinion. However, it definitely requires more scrutiny. Specifically, how is the convergence analysis different from mini-batch SGD analysis with bounded variance and smooth and lipschitz continuous ? More specifically, what new thing do we learn here ? Also, sample coverage and occupancy computations are important to understand the dynamics of random sampling in the context of training. However, to get fair understanding of the contribution, you might want to compare against a mini-batch SGD and talk about number of batches vs occupancy / coverage. (instead of epochs)**
>
> **A1:** The convergence analysis is different from the conventional mini-batch SGD analysis in that every epoch the stochastic gradients are only sampled from **a subset of the full training data**. However, since the subset selection is uniform at random and with replacement, the expectation taken over both the indices and subsets is the same as the conventional analysis where the expectation is only taken over the indices.  That's the difference and that is the reason that we require the random sampling to be with replacement. In other words, with replacement we can guarantee that the gradient is still unbiased.   If without replacement (e.g. just shuffling), the sampling is not independent anymore and the analysis would be a bit involved.  We are actually also investigating this scenario at the moment.
>
>
> **Q2: In the results, authors use same number of epochs across different methods. However, when sampling, the number of epochs really loose their meaning. Instead, we can keep number of iterations the same. Because it is difficult to compare between, say, 10% and 20% sampling rate if the 20% is runs twice as many iterations.**
>
> **A2:** The reason we conduct soft sampling in terms of epochs is that we want to make a fair comparison with GRAD-MATCH which is carried out in epochs.  In order to make this comparison fair, we follow the implementation of GRAD-MATCH and only replaced the OMP-based coreset selection with randomized subset selection.  That being said, we also ran pilot experiments with more epochs when a large fraction of data is used.  Our results show that the difference in terms of word error rate in payload is not obvious.  After enough epochs, the word error rates will plateau.
>
> **Q3. The argument that random sampling is competent for a lot of cases is a well accepted belief in the community. I am not sure if there has been a extensive study on the topic though. However, it should be noted that there will be cases where coresets will strongly outperform random sampling w.r.t accuracy and showing that random sampling beats coresets on a handful of datasets does not support using random sampling is better than coresets for training. In order to make a strong argument, i would suggest the following two directions - a. Use a lot of real datasets from different domains with learning and do an extensive study confirming what people believe. Having such a study would be very useful. b. Else, maybe you want to make an argument in favor of random sampling even in adversarial cases. In such a case, hand design data distributions where random sampling is bad (so in terms of #iterations, coresets beat random sampling) . But in training scenarios, using the advantage of fast sampling random sampling is competent with coresets in terms of #seconds to converge.**
>
> **A3:** Thanks for the suggestion.  These are valid points.  We will investigate these directions.
>
> **Q4. Also, it is not clear what the novelty of the paper is specifically because random sampling for training is the norm. It would be useful if authors would elaborate on the novelty. One interesting direction is ... Your approach is then somewhere in the middle, where you select a subset (which obviously has no repetitions) and then make batches from the subset. However the two different subset selections can have overlaps. You can analyse effect of this granularity that you introduce w.r.t occupancy, coverage and variance of gradient estimate for convergence (in theory and simulations) and in training convergence performance (# iterations required to converge).**
>
> **A4:** In this work we revisit randomized subset selection as a powerful technique for efficient large-scale training when dealing with massive amount of data, rather than being treated as an under-performing baseline in most of the existing literature.  To that end, we analyzed its convergence behavior and sampling dynamics which, to the best of our knowledge, are new.  Also, we think the reviewer's suggestion on the sampling granularity is legitimate and insightful. We will consider it in our future work.

---

### Official Review · Reviewer_u569 · 2022-10-25

**Confidence:** 4
**Clarity, Quality, Novelty And Reproducibility:** The paper is well written and easy to…
**Correctness:** 3
**Technical Novelty And Significance:** 1
**Empirical Novelty And Significance:** 2
**Recommendation:** 3

**Strength And Weaknesses:**

Strength:
1. The theoretical analysis of convergence and coverage of the sampling method could be a useful supplement to the methodology.

Weaknesses:
1. There is no major technical contribution in this paper. Algorithm 1 seems to be a standard practice of DNN training.
2. The theoretical results of convergence might be useful. It is not clear how the analysis coverage and occupation could be useful.
3. The evaluation part is weak. A larger scale dataset, like ImageNet is required for evaluation for validating the effectiveness.

**Summary Of The Paper:**

This paper analyze the soft sampling method for training DNNs, which selects a subset uniformly at random with replacement from the full data set in each epoch. Analysis of convergence rate and coverage are conducted.

**Summary Of The Review:**

As the technical contribution of this paper is weak, I recommend reject.

---

> ### Author Response · Authors · 2022-11-17
> **response to reviewer u569**
>
> **Q1:  There is no major technical contribution in this paper. Algorithm 1 seems to be a standard practice of DNN training.**
>
> **A1:** Please kindly note that Algorithm 1 is used to illustrate how to embed random subset selection in the traditional practice of DNN training which mostly uses the full training set. We do not intend to claim novelty in this algorithm.  This is also used to compare with the GRAD-MATCH algorithm where the random subset selection is replaced with OMP-based coreset selection. Other than the subset selection part, it is the standard practice of DNN training.
>
>
> **Q2: The theoretical results of convergence might be useful. It is not clear how the analysis coverage and occupation could be useful.**
>
> **A2:** Coverage and occupation would help users gain insight into the soft sampling technique and have a better understanding of the behavior of the algorithm.  More importantly, it will guide the users' experimental designs. For instance, given a pre-defined subset cardinality (e.g. 10% of data), a user can choose an appropriate number of epochs for a desired data coverage (e.g. 20 epochs or 30 epochs). Once the number of epochs is selected, the user can design the learning rate schedule accordingly. To the best of our knowledge, this is the first theoretical efficiency justification of determining such training hyper-parameters.
>
> **Q3: The evaluation part is weak. A larger scale dataset, like ImageNet is required for evaluation for validating the effectiveness.**
>
> **A3:** Please note that the payload dataset is a much larger dataset than ImageNet! The difference is that it is from the speech recognition domain instead of computer vision.  Our experimental design tried to cover different domains (image classification and speech recognition) and various dataset sizes (CIFAR10, a small dataset; Librispeech, a medium dataset; and Payload, a large dataset). We may need to validate the proposed technique on more datasets, but claiming that our experiments do not use a large-scale dataset is incorrect.

---

### Decision · Program_Chairs · 2023-01-20

**Decision:**

Reject

**Justification For Why Not Higher Score:**

Lack of novelty and insights.

**Justification For Why Not Lower Score:**

N/A

**Metareview: Summary, Strengths And Weaknesses:**

The paper proposed efficient training of neural networks by selecting a subset uniformly at random with replacement from the full data set in each epoch. The authors derive a theoretical convergence guarantee for the proposed random sampling method on non-convex objective functions and provide convergence rates. They also evaluate the performance of the proposed method through experiments on a few datasets. I do not see the novelty of the proposed method to be significant, which is in-line with the reviewers' comment. Besides, it would be more useful if the authors evaluate the performance of their method on datasets with different characteristics (e.g. imbalanced classes), and characterize the use-cases where it works and the use-cases where it does not work. Hence, I do not recommend acceptance.